# Prediction of Vascular Access Stenosis by Lightweight Convolutional Neural Network Using Blood Flow Sound Signals

**DOI:** 10.3390/s24185922

**Published:** 2024-09-12

**Authors:** Jia-Jung Wang, Alok Kumar Sharma, Shing-Hong Liu, Hangliang Zhang, Wenxi Chen, Thung-Lip Lee

**Affiliations:** 1Department of Biomedical Engineering, I-Shou University, Kaohsiung 82445, Taiwan; wangjj@isu.edu.tw (J.-J.W.); a342824050@gmail.com (H.Z.); 2Department of Computer Science and Information Engineering, Chaoyang University of Technology, Taichung 413310, Taiwan; 3Division of Information Systems, School of Computer Science and Engineering, The University of Aizu, Aizu-Wakamatsu City 965-8580, Fukushima, Japan; wenxi@u-aizu.ac.jp; 4Department of Cardiology, E-Da Hospital, Kaohsiung 84001, Taiwan; lip1969@hotmail.com

**Keywords:** hemodialysis, vascular access, fistula, deep learning, lightweight convolutional neural network (CNN)

## Abstract

This research examines the application of non-invasive acoustic analysis for detecting obstructions in vascular access (fistulas) used by kidney dialysis patients. Obstructions in these fistulas can interrupt essential dialysis treatment. In this study, we utilized a condenser microphone to capture the blood flow sounds before and after angioplasty surgery, analyzing 3819 sound samples from 119 dialysis patients. These sound signals were transformed into spectrogram images to classify obstructed and unobstructed vascular accesses, that is fistula conditions before and after the angioplasty procedure. A novel lightweight two-dimension convolutional neural network (CNN) was developed and benchmarked against pretrained CNN models such as ResNet50 and VGG16. The proposed model achieved a prediction accuracy of 100%, surpassing the ResNet50 and VGG16 models, which recorded 99% and 95% accuracy, respectively. Additionally, the study highlighted the significantly smaller memory size of the proposed model (2.37 MB) compared to ResNet50 (91.3 MB) and VGG16 (57.9 MB), suggesting its suitability for edge computing environments. This study underscores the efficacy of diverse deep-learning approaches in the obstructed detection of dialysis fistulas, presenting a scalable solution that combines high accuracy with reduced computational demands.

## 1. Introduction

It is essential for patients with end-stage renal disease (ESRD) to undergo hemodialysis (HD) treatment two or three times a week [1]. These ESRD patients need to perform the operation of certain vascular access, when their veins could not be used to accomplish the regular HD procedure. However, following successful vascular access, stenosis or obstruction will eventually reoccur after prolonged use. Consequently, HD treatment cannot be performed efficiently. Hence, the vascular access needs to be monitored to detect the degree of stenosis or obstruction in each HD treatment. In cases where the vascular access demonstrates stenotic or thrombotic phenomena, physicians will perform repairs needed to maintain functionality of the hemodialysis treatment. Therefore, vascular access can be considered a second lifeline for ESRD patients.

In general, vascular access used for HD can be categorized into two main types: autogenous (native arteriovenous fistula, AVF) and prosthetic (arteriovenous graft, AVG) [2]. AVF primarily uses the radial cephalic fistula and the brachiocephalic fistula, while AVGs are either looped or straight. Typically, looped grafts are used to connect the brachial artery to the basilar vein, and straight grafts connect the radial artery to the basilar vein. The ideal vascular access is the autogenous venous fistula, due to its lower complication rates and longer patency compared to prosthetic grafts [3,4]. With proper maintenance, there is a one-third chance that prosthetic grafts will last more than three years, and autogenous fistulas can last more than ten years.

The number of patients with kidney diseases is rising in several countries. Notably, Taiwan has the highest prevalence of ESRD globally, surpassing both the general prevalence (the proportion of the population affected by the disease) and its incidence rate (the number of new patients relative to the total population) [5,6]. Furthermore, prior research indicates that the prevalence of chronic kidney disease (CKD) in Taiwan is alarmingly high at 11.3%, meaning nearly 1 in every 10 individuals suffers from CKD [7,8,9]. Statistics from the National Health Insurance Agency reveal that over 85,000 people were undergoing kidney dialysis in Taiwan in 2020, with numbers continuing to rise. Consequently, Taiwan is often referred to as a “kidney dialysis island”, with a prevalence rate of over 3772 dialysis patients per million people, the highest ESRD prevalence rate treated globally, as shown in Figure 1 [5]. Additionally, Taiwan sees over 470 new diagnoses of ESRD per million people each year, the fastest rate worldwide [10].

Vascular phonoangiography (PAG) is a convenient technique to register the blood flow sounds through auscultation. The recorded sound signals are analyzed with the time- and frequency-domain characteristics to evaluate the conditions of vascular access [11,12,13,14,15]. Moreover, various nonlinear analysis techniques have been applied to examine the irregularities in PAG signals of vascular access under both obstructed and unobstructed conditions [16]. Ota et al. developed a method to detect and classify arteriovenous fistula stenosis severity using the fistula’s sounds, converted into spectrograms for analysis with Convolutional Neural Network (CNN) and CNN plus Gate Recursive Unit (GRU)/Long Short-Term Memory (LSTM) models [17]. While the method shows promising results in a five-item classification test, its accuracy on new data is still low. Julkaew et al. [18] employed adaptive deep learning (DL) techniques to predict the quality of vascular access in patients undergoing hemodialysis. Peralta et al. [19] applied the XGBoost algorithm to predict the failure of AVF within three months among patients receiving dialysis in-center. Wang et al. [20] employed a DL model for diagnosing renal artery stenosis, using multimodal fusion of ultrasound images, spectral waveforms, and clinical data. Li and He [21] introduced a neural network (NN) algorithm designed for diagnosing arterial stenosis. Nguyen et al. [22] employed a 1D-CNN to assess signal quality for effectively predicting real-time blood flow volume. Chung et al. [23] employed deep learning to predict arteriovenous access dysfunction using spectrogram images. Furthermore, previous studies have used machine learning models to predict CKD [24,25,26,27].

In Taiwan, the rising prevalence of kidney diseases requiring dialysis poses significant challenges to the healthcare system and patient welfare. Prompt identification and treatment of vascular access complicated by stenosis or obstruction are crucial for prolonging their functionality and ensuring patient safety. Traditionally, clinicians use ultrasound imaging to detect changes in the blood flow of vascular accesses, a method limited by its subjectivity and potential for varied interpretations among healthcare providers [28]. Monitoring the venous blood pressure and blood flow of vascular access could increase the effect of preventive repair on obstructed rate in the impaired AVF, but not increase the sensitivity in the AVG [29]. The timely detection of vascular accesses in the thrombosis rate not only facilitates the necessary hemodialysis, but also extends the fistula’s lifespan.

The aim of this study is to explore the potential and effectiveness of non-invasive acoustic analysis for identifying stenoses or blockages within the vascular accesses (AVF or AVG). Utilizing a condenser microphone to record blood-flow sounds before and after angioplasty, this study intends to analyze these sounds to distinguish between obstructed and unobstructed blood flows in the vascular access. We collected the PAG signals of 119 patients before and after the angioplasty surgery. The PAG signals used the short-time Fourier transform (STFT) to obtain the spectrogram images, time-frequency images. By analyzing 3819 sound samples, it assesses the proposed lightweight two-dimension CNN model alongside pretrained deep learning (DL) models, including VGG16 and ResNet50, for their accuracy in classifying the stenosis and non-blockage of vascular access. The lightweight CNN model demonstrated the best performance, which is suitable for edge computing.

## 2. Materials and Methods

The flowchart of this study for identifying whether the stenosis or obstruction of vascular access or not with the blood-flow sound is illustrated in Figure 2. The blood-flow sound signals were collected from patients at the Cardiovascular Center of the E-Da Hospital. The physician identified the vascular access of patients with the stenosis condition, and performed the angioplasty surgery to clear the obstructed blood-flow pathway. Then, the blood-flow sounds are measured before and after the surgery, which are converted into spectrogram images to visualize time-frequency components. In data preprocessing, the lower frequency components represent the more important features. Thus, the raw spectrogram images are filtered the components in the higher frequency. The dataset is split into 80% for training and validation and 20% for testing. The proposed lightweight CNN model, and pretrained VGG16 and Resnet50 models are trained using the training subset, with the validation subset preventing overfitting. The models’ performances are then evaluated using the testing subset, ensuring accuracy and reliability in the classification of blood-flow sounds.

### 2.1. Data Source

This clinical trial was approved by the Institutional Review Board of the E-Da Hospital, Kaohsiung, Taiwan (No. EMRP-107-142), and informed consent was acquired from each participant prior to initiation of the study. In the informed consent, the physician explained the materials and measurement procedure of this experiment. Moreover, the blood-flow sound measurement was the noninvasive measurement.

The blood-flow sounds through the fistula are measured with a stethoscope at three points, as outlined in Figure 3: at the entrance position (A), midpoint (B), and exit position (C) of the blood flow. Approximately 30 min before fistula clearing surgery, a nurse working in Cardiovascular Center identified these points on the patient for measurement. For each point, the blood-flow sound was measured for 20 s. Following the surgery, and after a 2-min rest period for the patient, the measurements were repeated at the same points for 20 s each. All measurements were completed within a 5-min period. Once the initial measurement points were inaccessible post-operation, the nearest feasible points were chosen.

Data collection was conducted using the BIOPAC MP-150 (BioPack Systems Inc., Goleta, CA, USA) system for physiological signal acquisition, both before and after the angioplasty surgery, with each patient’s data distinctly stored on a computer hard drive. The AcqKnowledge software was then applied to filter, select and archive these signals into Excel files for analysis. A self-made stethoscope combined with a condenser microphone was used to measure the blood-flow sound.

The heart rate typically falls within low frequencies, with measurements at this institute concentrating between 1–3 Hz. However, the human heart sound can be detected at frequencies ranging from 20 Hz to 10 kHz. Thus, the sampling frequency was set at 20 kHz. AcqKnowledge software was employed for the signal processing, including baseline wounding and noise, to standardize the data. Given the variability in cardiac cycles among patients due to factors such as age and illness, the signal of blood-flow sound was segmented into uniform 3-s durations to ensure signal with at least one heat beat cycle. These data allowed for comparison of blood-flow signals before the surgery (obstructed fistula) and after the surgery (unobstructed fistula).

This study conducted at the Cardiovascular Center of the E-Da Hospital, Kaohsiung, Taiwan, compiled 119 patient records. The dataset included 7 cases lacking postoperative data, 98 cases of successful treatment for overcoming the stenosis or fistula obstruction, 1 case of surgical failure, and 3 indeterminate cases due to issues like improper instrument adjustment or uncertain A, B, or C point locations. Hence, 106 datasets were available belonging to the stenosis or obstructed case, and 105 belonging to the unobstructed case. The blood-flow sounds measured at points A, B, and C were segmented with 3 s. This approach yielded 1923 samples for the obstructed class, and 1896 samples for the unobstructed class.

### 2.2. Data Preprocessing

The STFT is introduced as a method to address the inherent limitations of the fast Fourier transform (FFT), particularly in the analysis of non-stationary or noisy signals. STFT is predominantly utilized for the extraction of narrow-band frequency components within such signals [30]. The underlying concept involves segmenting the original signal into small, localized time windows, followed by the application of FFT to each segment. This approach enables the representation of temporal variations in the signal’s frequency content, effectively capturing the dynamic behavior within each time window [31]. In the data preprocessing phase, we first used MATLAB (The MathWorks Inc., Natick, MA, USA) to apply the STFT to transform the signal of blood-flow sound. We used a window size of 256, with 50% overlap of the window size, and set the number of FFT points to 1024. This procedure can increase the resolution in the frequency domain. The STFT provided detailed information on the time and frequency domains, which could be used to classify the obstructed and unobstructed classes, without compromising its intrinsic characteristics. Moreover, this transformation was crucial for converting time-domain signals into a format that can be analyzed more effectively in the frequency domain. Figure 4a,b show the spectrogram images of blood-flow sound before and after angioplasty surgery. We can find their difference happening on lower frequencies. Thus, this study filtered the high frequency of spectrogram from 2001 Hz to 100,000 Hz, as the reduced spectrogram images encompassed the most relevant information, as shown in Figure 4c,d. The choice of the 0–2000 Hz frequency range in the STFT was to capture relevant physiological signals related to blood-flow sound. This narrow band allows the model to learn from a more comprehensive spectrum, potentially enhancing its ability to detect various patterns associated with different abnormalities. This filtering step could be essential to eliminate noise and irrelevant frequency components, ensuring that the samples fed into the model is of high accuracy. Moreover, the image size was reduced to 110 × 110 pixels.

### 2.3. Dataset Partitions

To train and test the model, the dataset must be divided. Table 1 shows the dataset partitions. The training dataset, making up 80% of the total, includes both the training and validation sets. The training set is used to fit the model, aiming for generalization, while the validation set, 20% of the training data, helps in model verification and parameter tuning. We ensure consistent data partitioning across experiments by fixing the random seed with the scikit-learn library. The test dataset, the remaining 20%, evaluates the model’s accuracy on new data. Only the training and validation sets undergo data enhancement to improve model fit without affecting the authenticity of the test data, which includes 765 samples, with 385 for obstructed samples and 380 for unobstructed samples.

### 2.4. Model Architecture

In this study, we proposed lightweight two-dimension (2D) CNN, as shown in Figure 5. This experiment employs TensorFlow’s Sequential API to construct a portion of the Deep Neural Network (DNN). The lightweight CNN model processes 110 × 110 pixels input spectrogram images through a series of layers designed for efficient feature extraction and classification. It starts with a 2D convolutional layer with 4 filters, followed by batch normalization to stabilize training. A depthwise separable convolution layer and max-pooling for the down sample of feature maps, which is repeated with an additional 2D convolutional layer with 8 filters. The feature maps are flattened and passed through a fully connected (FC) layer with 32 units. To prevent overfitting, a dropout layer with a 0.5 rate is applied before reaching the final output layer (only one node), making the model robust and effective for image classification tasks. In this study, we employed a learning rate of 0.0001 to ensure gradual and stable convergence during training. The ReLU (Rectified Linear Unit) activation function was utilized to introduce non-linearity, enabling the model to learn complex patterns. Additionally, a kernel size of 3 was defined, meaning the convolutional filters applied during operations will have dimensions of 3 × 3, which is effective in capturing local features from the input data.

Many models [32], pre-trained on the ImageNet database, are readily available for use with Keras and TensorFlow. This study employed two pre-trained models, VGG16 and ResNet50, and compared them with the proposed model.

### 2.5. Loss Function

In each instance of training, we quantify the discrepancy between the predicted and actual data. This quantification serves as a crucial indicator of the model’s efficacy, encapsulated by the loss function. The present investigation adopts a binary classification framework, wherein the output layer’s activation function is delineated by a Sigmoid curve. Consequently, the output values are confined within the interval [0, 1]. Given this constraint, the binary cross-entropy function emerges as the appropriate choice for evaluating model performance [33]. The following is the formula for binary cross entropy:(1)$$Loss=-\frac{1}{N}\sum_{i=1}^{N}y_{i}\log\left(p\left({\hat{y}}_{i}\right)\right)+\left({1-y}_{i}\right)\log\left(1-p\left({\hat{y}}_{i}\right)\right)$$

As a loss function for binary classification problems, where *y_i_* represents the binary label (0 or 1), and $p({\hat{y}}_{i})$ denotes the probability that the output correctly matches the label. It can be observed from Equation (1) that when the label *y_i_* is 1, and the predicted value $p({\hat{y}}_{i})$ approaches 1, the *Loss* = −log(*p*(${\hat{y}}_{i}$)) decreases, nearing 0. Similarly, when the label *y_i_* is 0, and the predicted value $p({\hat{y}}_{i})$ approaches 0, the *Loss* = −log(1 − *p*(${\hat{y}}_{i}$)) also decreases, approaching 0. Subsequently, the average loss across *N* output values is computed to determine the overall loss for the dataset.

### 2.6. Evaluation Metrics

This study analyzed the model’s performance using precision, recall, specificity, negative predictive value, and accuracy [34]. In the equations below, ‘*TP*’, ‘*TN*’, ‘*FP*’, and ‘*FN*’ are used to represent ‘True Positive’ (accurate positive detections), ‘True Negative’ (correct negative classifications), ‘False Positive’ (incorrect positive classifications), and ‘False Negative’ (incorrect negative classifications), respectively.

The *Precision* represents the proportion of correctly classified positive predictions out of all positive predictions.
(2)$$Precision=\frac{TP}{TP+FP}$$

The *Recall* represents the proportion of positive samples that were correctly predicted, compared to the total number of actual positive samples.
(3)$$Recall=\frac{TP}{TP+FN}$$

The *F*1-score is an accuracy measure that balances both precision and recall, providing a single comprehensive metric.
(4)F1-score=2×(Precision×Recall)/(Precision+Recall)

*Accuracy* measures the proportion of all predictions (both positive and negative) that a model correctly identifies out of the total predictions made.
(5)$$Accuracy=\frac{TP+TN}{TP+TN+FP+FN}$$

## 3. Experimental Results

### 3.1. Classification Result of CNN Models

The classification performances of three different CNN models using spectrogram images with the full band are provided in Table 2. In a two-class classification task, VGG16 model achieved an overall accuracy of 0.53, with strong recall (0.88) for obstructed vascular accesses but weaker performance for the unblocked class, reflected in lower precision (0.58) and recall (0.17). The ResNet50 model performed better overall, with a 0.59 accuracy, showing more balanced precision and recall for both classes, leading to a more consistent *F*1-score. The lightweight CNN model demonstrates a classification accuracy of 0.55 for the obstructed class, with precision and recall values of 0.58 and 0.38, respectively, resulting in an *F*1-score of 0.46. These metrics indicate moderate precision but low recall, meaning the model is accurate in predicting obstructed class but misses many actual obstructed cases. The unobstructed class achieved a recall of 0.72, higher than its precision measure of 0.53 and *F*1-score of 0.61. The higher precision suggests that when the model predicts the unobstructed class, it is more often correct; better recall means it is good at identifying actual cases of unobstructed class.

This can also be understood in more detail from the confusion matrixes of three CNN models shown in Figure 6. The VGG16 model correctly predicted 339 obstructed samples and made 46 incorrect predictions. It also correctly identified 63 unobstructed samples while misclassifying 317, as shown in Figure 6a. The ResNet50 model had 237 correct predictions and 148 errors for obstructed samples, and 213 correct predictions with 167 misclassifications for unobstructed samples, as shown in Figure 6b. The lightweight CNN model correctly predicted 146 obstructed samples but made 239 incorrect predictions. Conversely, the unobstructed samples were correctly predicted 273 times and misclassified 107 times, as shown in Figure 6c. This imbalance indicates that the model struggles to achieve optimal separation between the two classes, particularly in consistently identifying obstructed class, which corresponds with the observed lower recall for the obstructed class.

Table 3 presents a comparative performance evaluation of three CNN models—VGG16, ResNet50, and lightweight CNN—using filtered spectrogram images. The performance metrics evaluated include accuracy, precision, recall, and *F*1-score for both obstructed and unobstructed classes. Performance evaluation against the accuracy, precision, recall, and *F*1-score for both obstructed and unobstructed classes is carried out by each model. The VGG16 model scored 0.95 accuracy, 0.96 precision, 0.95 recall, and 0.95 *F*1-score for the obstructed class, overall almost the same for the unobstructed class, which was accompanied by a well-balanced performance of this model. The ResNet50 model showed better performance with an overall accuracy of 0.99. It has a perfect precision of 1.00, a recall of 0.97, an *F*1-score of 0.99 for the obstructed class, and very high performances for the unobstructed class. The other two performing models are compared to it: lightweight CNN model, which achieved an overall perfect accuracy of 1.00, with the same result for precision, recall, and *F*1-score for both classes being at 1.00, meaning it also performed flawlessly; finally, Resnet50 achieved better results compared to VGG16, with accuracy up to 0.98 and *F*1-score up to 0.99 for the obstructed class. The confusion matrices in Figure 7 illustrate the performance of each model. Figure 7a, which represents the VGG16 model, shows some misclassifications but maintains high accuracy for both obstructed and unobstructed classes. Figure 7b, illustrating the ResNet50 model, demonstrates very few misclassifications, indicating highly reliable predictions suitable for real-world applications. Figure 7c, showing the lightweight CNN model, exhibits perfect performance with no misclassifications, accurately predicting all instances. This comprehensive evaluation underscores the superior accuracy and reliability of the lightweight CNN model, followed closely by ResNet50, with VGG16 performing robustly but slightly less impressively.

Figure 8 presents the training and validation performance metrics of three CNN architectures across 20 epochs. Figure 8a shows the training and validation accuracy of the VGG16 model, with a gradual increase and some fluctuations in validation accuracy. Figure 8b depicts the corresponding loss, showing a consistent downward trend with more variability in the validation loss. Figure 8c,d display the accuracy and loss for the ResNet50 model, respectively. Both results quickly stabilize at high accuracy and low loss values, indicating efficient learning and minimal error. Figure 8e,f present the accuracy and loss for the lightweight CNN model. The model also achieves near-perfect accuracy and minimal loss rapidly, demonstrating strong performance. VGG16 shows gradual improvement, while ResNet50 and lightweight CNN model quickly achieve high accuracy and low loss, indicating their robustness and efficiency.

### 3.2. Five-Fold Validation Classification Result

This study also used a five-fold validation method to analyze model performance. The five-fold cross-validation evaluation results for three models—VGG16, ResNet50, and lightweight CNN—are shown in Table 4, Table 5 and Table 6, respectively. Five-fold cross-validation is an essential technique in machine learning to measure the goodness of a model against tasks and inference, which enforces the reliability of the model. It reduces the problem of overfitting by training and validating the model on different data subsets, which makes it better in generalization performance on new data. This method provides robust performance by averaging metrics such as accuracy, precision, recall, and *F*1-score over five runs. It also maximizes data utilization, which is critical in medical research where data can be limited.

Table 4 highlights VGG16’s performance, showing fold accuracies between 0.96 and 0.99, with an overall average accuracy of 0.98. The precision, recall, and *F*1-score also average 0.98, indicating that VGG16 consistently delivers reliable performance, albeit with slight variations across different folds. Table 5 details evaluating results of ResNet50, showing perfect scores in all metrics across all five folds. With an average accuracy, precision, recall, and *F*1-score of 1.00, ResNet50 demonstrates exceptional performance and robustness, proving to be a highly effective model for the task. Table 6 presents the results for lightweight CNN model, which also achieves perfect scores in results matrix for each fold. With an average accuracy, precision, recall, and *F*1-score of 1.00, lightweight CNN model matches ResNet50’s performance while likely benefiting from reduced computational and storage demands.

### 3.3. Model Size

In edge computing for medical applications, selecting an appropriate deep learning model is essential due to limited computational resources and storage capacities. Table 7 compares VGG16, ResNet50, and lightweight CNN model based on their parameters and memory sizes. VGG16, with 14.86 million parameters and a memory size of 57.9 MB, offers simplicity and depth but may be too large for resource-constrained devices. ResNet50, featuring residual connections, has 23.66 million parameters and a memory size of 91.3 MB, providing high accuracy but requiring more computational resources. In contrast, the lightweight CNN model, with only 201,378 parameters and a memory size of 2.37 MB, is highly efficient. It balances accuracy with reduced computational and storage demands, making it ideal for real-time medical applications on edge devices.

## 4. Discussion

This study aims to develop a lightweight CNN model for the non-invasive detection of vascular access stenosis using blood-flow sounds, providing a novel approach to enhancing patient care in patients undergoing hemodialysis. The proposed model exhibits rigorously compared with established CNN models, specifically VGG16 and ResNet50. To classify the obstructed and unobstructed fistula, signals of blood-flow sound are transformed into spectrogram images, a critical step that enables the extraction of relevant features from the time and frequency domains. In this study, we used spectrogram images within a narrow bandwidth below 2000 Hz, because we observed that the low-frequency band carries more significant information related to blood-flow sounds. This finding aligns with prior research, as the selected frequency range was informed by earlier studies [35]. In Table 2 and Table 3, the results metrics with the full band (0–10,000 Hz) are lower than metrics with the narrow band (0–2000 Hz). Notably, the performances of all models were consistently higher when using the spectrogram images with the narrow band. This finding also showed that the bandwidth of blood flow sound would be within the low-frequency band. Filtering out higher frequencies not only reduced noise but also emphasized relevant features, and enhanced the model’s performance. Although the number of samples in Table 2 is small, the features of obstructed and unobstructed classes are very different. Thus, in Table 4, Table 5 and Table 6, the models with the five-fold cross-validation exhibit values indicative of high robustness and performance. The results, as presented in Table 3, demonstrate that the lightweight CNN model achieves an unparalleled prediction accuracy of 100%, significantly outperforming ResNet50 and VGG16, which record accuracies of 99% and 95%, respectively. This superior performance underscores the efficacy of the lightweight CNN model in accurately identifying vascular conditions.

Mansy et al. studied 11 patients whose vascular sounds were recorded before and after angiography. They subsequently found changes in acoustic amplitude and/or spectral energy distribution. Certain acoustic parameters with the change in the degree of stenosis had a high correlation coefficient, approaching 0.98 [36]. Sung et al. proposed a multivariate Gaussian distribution (MGD) model to detect significant vascular access stenosis with the phonoangiographic method. The auditory spectrum flux and auditory spectral centroid were extracted from the blood-flow sound of 16 HD patients with AVG. Its accuracy only reached 83.9% [12]. In this study, we complied 119 patient records. There were 106 datasets available belonging to the stenosis or obstructed case, and 105 datasets belonging to the unobstructed case. The signal was transferred to a spectrogram image via STFT. In order to enhance the feature of spectrogram image, the filtered spectrogram image was used to classify the obstructed and unobstructed classes via the proposed lightweight CNN model, subsequently showing an accuracy of 100%. This result shows that the two-dimension CNN model bears the potential benefit of detecting vascular access stenosis.

To assess the suitability of the model for edge computing applications, this study also analyzes the memory size. The lightweight CNN model is identified as having the smallest memory size compared to VGG16 and ResNet50, making it especially suitable for use in environments with limited resources. Its compact memory size allows it to function efficiently on devices with restricted computational power and storage capacity, such as mobile devices or portable medical equipment. The memory size of a model is a crucial aspect of edge computing in healthcare, as it determines the feasibility of deploying advanced algorithms on resource-constrained devices [37]. This efficiency is crucial in real-time medical applications where rapid and accurate predictions are necessary for timely interventions.

This study also compares our results with a previous edge computing study by Zhou et al. [38], where they achieved a specificity value of 0.791. In contrast, our study demonstrates a specificity value exceeding 95% across all employed models. Moreover, another study by Ota et al. [17] achieved an accuracy of 0.821, precision of 0.414, and recall of 0.578, whereas our study outperforms these results. This indicates a significant improvement in the ability to detect the vascular access stenosis. Table 8 summarizes various studies that employed deep learning models to analyze AVF sounds and clinical data for predicting AVF functionality and vascular access quality in HD patients. The dataset, model type, methods used, and key outcomes are outlined for each study, highlighting the performance and predictive accuracy of the models.

Overall, the findings from this study highlight the potential of the lightweight CNN model not only in providing high accuracy for the detection of vascular access stenosis but also in offering a practical solution for edge computing scenarios. The combination of high predictive accuracy, robustness through cross-validation, and efficient memory size positions the lightweight CNN model as a promising model for improving the diagnosis and management of vascular access conditions in dialysis patients.

In the future, collecting more patient data will improve the generalizability of DL models, potentially reducing signal-processing needs. Utilizing microcontrollers for data collection and prediction could save space and memory, lowering medical application costs. Enhanced processing of input features, like using Fast Fourier Transform for more detailed time series signal preprocessing, may maintain computation levels while increasing prediction accuracy.

This study has several limitations. The dataset, while robust, may not fully capture the diversity of patient conditions and blood-flow sound variations seen in one clinical practice. The transformation of audio signals into spectrogram images might introduce some information loss, affecting model performance. Additionally, the model’s real-world applicability requires further validation in diverse clinical settings. Future research should address these issues by using more varied datasets, exploring alternative data representations, and conducting extensive clinical testing.

## 5. Conclusions

A lightweight CNN model is proposed in this study to classify the stenosis of vascular access using blood-flow sounds, demonstrating its suitability for edge computing devices in the medical field. The model’s compact memory size is particularly advantageous for deployment in resource-constrained environments. Further experimentation involves transforming audio signals into spectrogram images using various processing methods to evaluate their effectiveness in deep learning models. The results indicate that the lightweight CNN model could be used to evaluate the fistula condition in dialysis patients in an objective and unsupervised manner, potentially enabling early detection and intervention for blood flow obstructions. The study sets the stage for further research and development in the application of artificial intelligence (AI) in healthcare diagnostics. The findings presented in this study help further enhance non-invasive diagnostic tools, showing how lightweight CNN models could be effectively employed in real-world clinical settings.

## Figures and Tables

**Figure 1 sensors-24-05922-f001:**
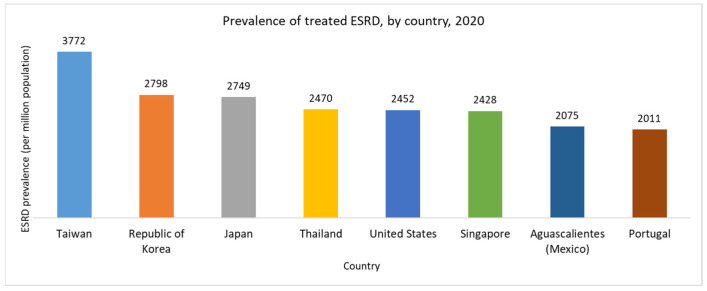
Prevalence rates of treated end-stage renal disease in several countries.

**Figure 2 sensors-24-05922-f002:**
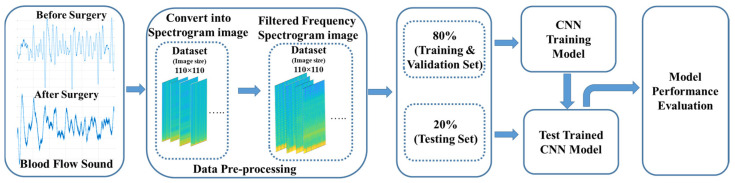
The flowchart of the overall study includes the blood-flow sound measurement, spectrogram image conversion, CNN models, and performance evaluation.

**Figure 3 sensors-24-05922-f003:**
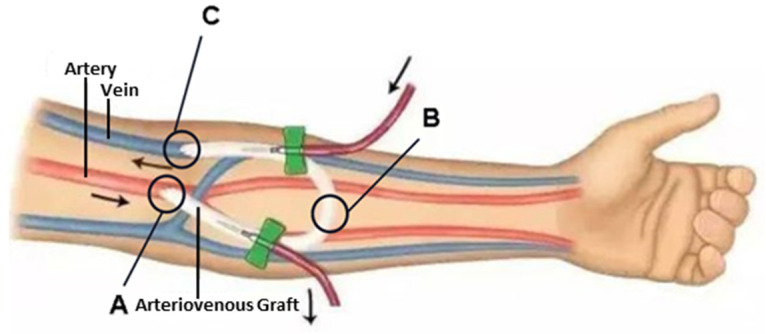
Three-point schematic diagram for measuring the blood-flow sound signals at the fistula. A is the entrance position of blood flow, B is the midpoint, and C is the exit position of blood flow.

**Figure 4 sensors-24-05922-f004:**
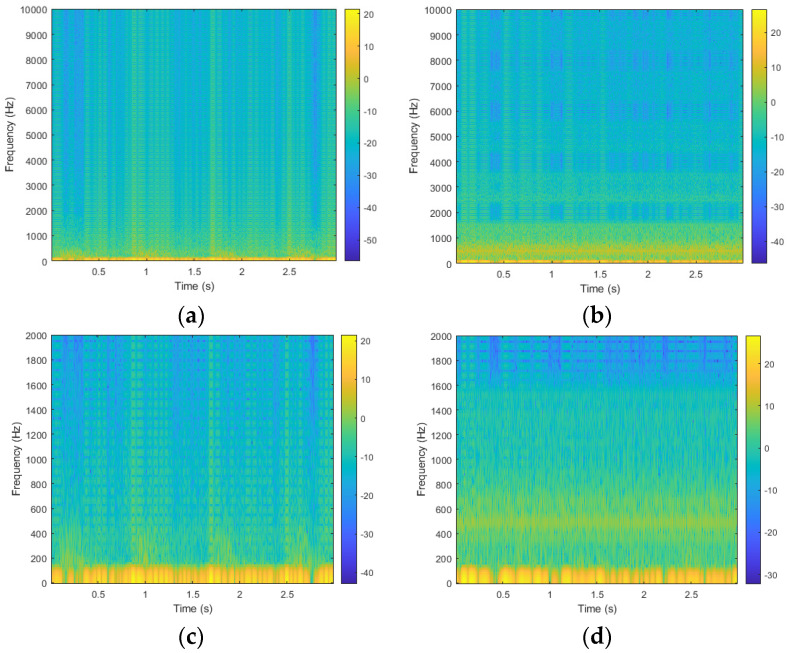
The spectrogram images before and after angioplasty surgery, (**a**) Spectrogram image before surgery; (**b**) Spectrogram image after surgery; (**c**) Filtered Spectrogram image before surgery; (**d**) Filter spectrogram image after surgery.

**Figure 5 sensors-24-05922-f005:**
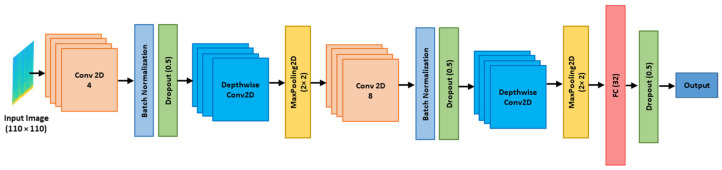
The architecture of proposed lightweight 2D CNN model. Input is the spectrogram image, and output is one node.

**Figure 6 sensors-24-05922-f006:**
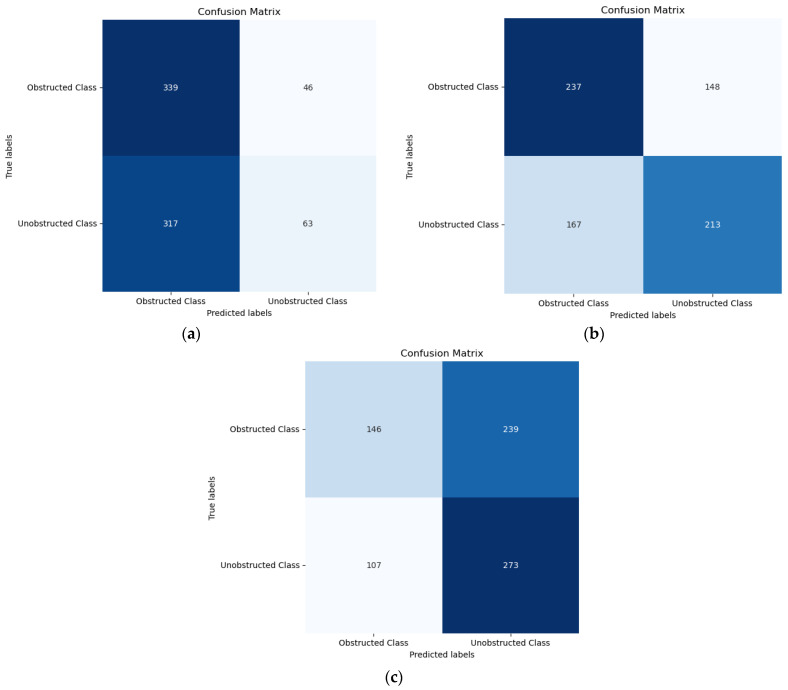
Confusion matrixes for the different CNN models using spectrogram images with the full band, (**a**) VGG16; (**b**) ResNet50; (**c**) Lightweight CNN.

**Figure 7 sensors-24-05922-f007:**
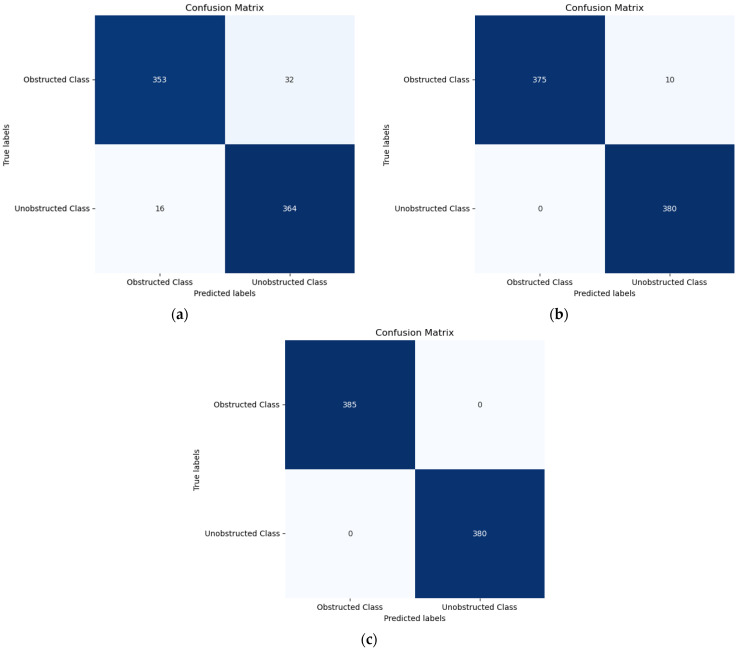
Confusion matrixes for the different CNN models using filtered spectrogram images, (**a**) VGG16; (**b**) ResNet50; (**c**) lightweight CNN.

**Figure 8 sensors-24-05922-f008:**
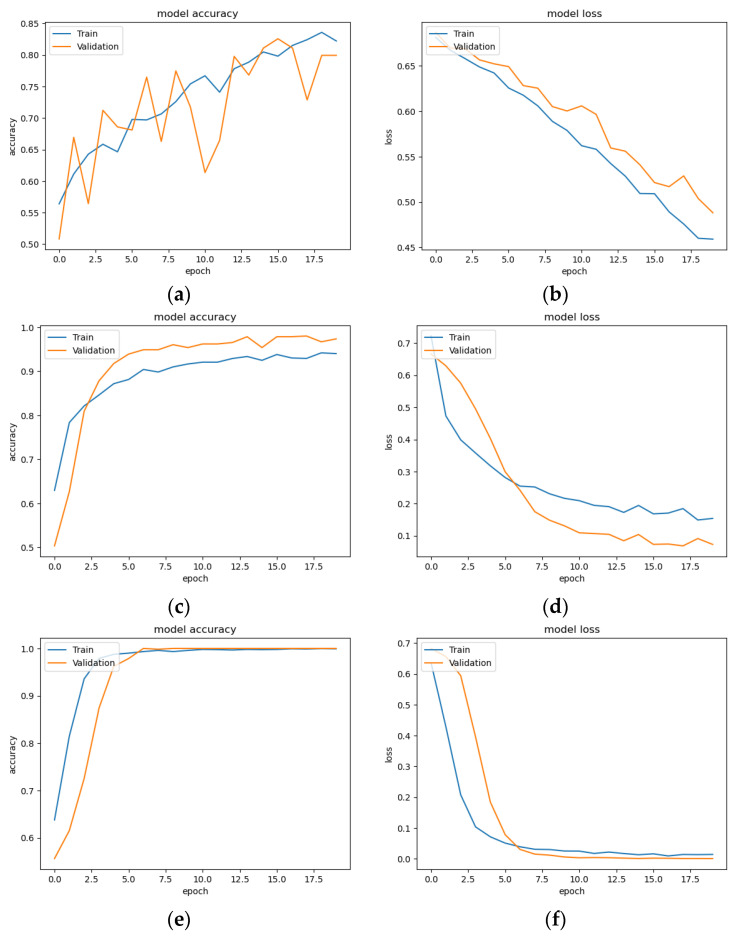
(**a**) Training and validation accuracy of VGG16; (**b**) Training and validation loss value of VGG16; (**c**) Training and validation accuracy of ResNet50; (**d**) Training and validation loss value of ResNet50; (**e**) Training and validation accuracy of lightweight CNN model; (**f**) Training and validation loss value of lightweight CNN.

**Table 1 sensors-24-05922-t001:** Dataset partitions for obstructed and unobstructed classes.

Dataset	Label	Train	Test	Total
Raw signal	Obstructed class	1538	385	1923
	Unobstructed class	1516	380	1896

**Table 2 sensors-24-05922-t002:** Evaluation results matrix for the VGG16, ResNet50 and lightweight CNN models using spectrogram images with the full band.

Models	Accuracy	Obstructed Class			Unobstructed Class		
		Precision	Recall	*F*1-Score	Precision	Recall	*F*1-Score
VGG16	0.53	0.52	0.88	0.65	0.58	0.17	0.26
ResNet50	0.59	0.59	0.62	0.60	0.59	0.56	0.57
Light-CNN	0.55	0.58	0.38	0.46	0.53	0.72	0.61

**Table 3 sensors-24-05922-t003:** Evaluation results matrix for VGG16, ResNet50 and lightweight CNN models using filtered spectrogram images.

Models	Accuracy	Obstructed Class			Unobstructed Class		
		Precision	Recall	*F*1-Score	Precision	Recall	*F*1-Score
VGG16	0.95	0.96	0.95	0.95	0.95	0.96	0.95
ResNet50	0.99	1.00	0.97	0.99	0.97	1.00	0.99
Lightweight CNN	1.00	1.00	1.00	1.00	1.00	1.00	1.00

**Table 4 sensors-24-05922-t004:** The results metric of five-fold evaluation for VGG16.

Fold	Accuracy	Precision	Recall	*F*1-Score
1	0.99	0.99	0.99	0.99
2	0.99	0.99	0.99	0.99
3	0.96	0.96	0.96	0.96
4	0.97	0.97	0.97	0.97
5	0.99	0.99	0.99	0.99
**Average**	**0.98**	**0.98**	**0.98**	**0.98**

**Table 5 sensors-24-05922-t005:** The results metric of five-fold evaluation for ResNet50.

Fold	Accuracy	Precision	Recall	*F*1-Score
1	1.00	1.00	1.00	1.00
2	1.00	1.00	1.00	1.00
3	1.00	1.00	1.00	1.00
4	1.00	1.00	1.00	1.00
5	1.00	1.00	1.00	1.00
**Average**	**1.00**	**1.00**	**1.00**	**1.00**

**Table 6 sensors-24-05922-t006:** The results metric of five-fold evaluation for lightweight CNN model.

Fold	Accuracy	Precision	Recall	*F*1-Score
1	1.00	1.00	1.00	1.00
2	1.00	1.00	1.00	1.00
3	1.00	1.00	1.00	1.00
4	1.00	1.00	1.00	1.00
5	1.00	1.00	1.00	1.00
**Average**	**1.00**	**1.00**	**1.00**	**1.00**

**Table 7 sensors-24-05922-t007:** Parameters and memory sizes of VGG16, ResNet50 and lightweight CNN model.

Models	Parameters	Memory Size
VGG16	14,862,242	57.9 MB
ResNet50	23,661,538	91.3 MB
Lightweight CNN	201,378	2.37 MB

**Table 8 sensors-24-05922-t008:** Summary of deep learning models for predicting AVF functionality and VA quality in HD patients.

Author	Dataset	Model	Method	Outcome
Ota et al. [17]	AVF sounds recording 1 min from 20 patients.	CNN + BiLSTM (Bidirectional Long Short-Term Memory)	Extracted heartbeat-specific arteriovenous fistula sounds.	0.70 to 0.93 of Accuracy, 0.75 to 0.92 of AUC.
Julkaew et al. [18]	Clinical data from 398 HD patients	DeepVAQ-CNN	Used Photoplethysmogram (PPG) to predict VA quality, trained and fine-tuned CNN.	0.92 of Accuracy, 0.96 of Specificity, 0.88 of Precision, 0.84 of *F*1-score, 0.86 of AUC.
Peralta et al. [19]	13,369 dialysis patients (EuCliD^®^ Database)	AVF-FM (XGBoost)	Predict AVF failure in 3 months.	0.80 of AUC (95% CI 0.79–0.81).
Nguyen et al. [22]	300 qualified and 202 unqualified PPG waveforms	1D-CNN + FCNN	Developed an ML algorithm to assess PPG signal quality for prediction of blood flow volume, using waveform quality criteria.	Transformed NN: 0.94 of Accuracy. 1D-CNN: 0.95 of Accuracy.
Zhou et al. [38]	2565 AVF blood flow sounds from 433 patients	Vision Transformer (ViT)	AVF sounds from 6 locations, pre-processed into Mel-spectrograms and recurrence plots.	ViT: 0.92 of Sensitivity, 0.79 of Specificity, 0.91 of *F*1-score.
Chung et al. [23]	437 audio recordings from 84 HD patients	CNN, ViT-GRU	AV access bruit recordings converted to Mel-spectrograms. Models trained to predict dysfunction.	CNN: 0.70 of F1 Score, 0.71 of AUC. ViT-GRU: 0.52 of *F*1-score, 0.60 of AUC.
Park et al. [39]	80 audio files from 40 HD patients.	ResNet50, EfficientNetB5, DenseNet201	Digital AVF sounds recorded, converted to mel spectrograms, and used DCNN models.	ResNet50: 0.99 of AUC. EfficientNetB5: 0.98 of AUC.

## Data Availability

Data are contained within the article.

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
