# Peer review of "Prediction of Vascular Access Stenosis by Lightweight Convolutional Neural Network Using Blood Flow Sound Signals"

_sensors, 2024, doi:10.3390/s24185922_

Round 1

Reviewer 1 Report

Comments and Suggestions for Authors

The authors proposed a lightweight CNN model to classify vascular access steno using blood flow sound signals, demonstrating a prediction accuracy of 100% against the ResNet50 and VGG16. However, the benefits and the novelty of the work are unclear to me. In contrast to Ultrasound imaging, it’s an invasive way of acquiring fistula blood flow sound signals as shown in Fig.3. Considering the proposed CNN model, the authors are expected to provide more state-of-the-art works to highlight its novelty.  

1.     Line 87-98, the reviewed work of AI in edge computing can be removed regarding that the description (line 70-86) already focused on CNN researches in detecting auscultatory sounds for predicting CKD.

Author Response

To Reviewer #1:

Thank the reviewer for his/her valuable comments that make better this manuscript. The texts in this revised manuscript have been corrected/ modified by red mark. It is our sincere hope that this revision will enhance readability and strengthen of the manuscript to satisfy the requirements of this prestigious journal.

Comments and Suggestions for Authors

  1. The authors proposed a lightweight CNN model to classify vascular access steno using blood flow sound signals, demonstrating a prediction accuracy of 100% against the ResNet50 and VGG16. However, the benefits and the novelty of the work are unclear to me. In contrast to Ultrasound imaging, it’s an invasive way of acquiring fistula blood flow sound signals as shown in Fig.3. Considering the proposed CNN model, the authors are expected to provide more state-of-the-art works to highlight its novelty.

ANS: Many thanks for reviewer’s comment. We modify the mentions in Introduction chapter to explain the aim of this study more clear, and in Methods chapter to provide the contribution of this study.  

Line: 97-108

The aim of this study is to explore the potential and effectiveness of non-invasive acoustic analysis for identifying the stenosis or blockage within the vascular accesses (AVF or AVG) used by kidney dialysis patients. Utilizing a condenser microphone to record blood-flow sounds before and after angioplasty, the study intends to analyze these sounds to distinguish between obstructed and unobstructed blood flows in the vascular access. We collected the PAG signals of 119 patients before and after the angioplasty surgery. The PAG signals used the short-time Fourier transform (STFT) to get the spectrogram images, time-frequency images. By analyzing 3,819 sound samples, it assesses the proposed lightweight two-dimension CNN model alongside pretrained deep learning (DL) models, including VGG16 and ResNet50, for their accuracy in classifying the stenosis and non-blockage of vascular access. The lightweight CNN model had the best performance, which is suitable for edge computing.

Line: 109-123

  1. Materials and Methods

The flowchart of this study for identifying whether the stenosis or obstruction of vascular access or not with the blood-flow sound is illustrated in Figure 2. The blood-flow sound signals were collected from patients at the Cardiovascular Center of the E-Da Hospital. The physician identified the vascular access of patients with the stenosis condition, and performed the angioplasty surgery to clear the obstructed blood-flow pathway. Then, the blood-flow sounds are measured before and after the surgery, which are converted into spectrogram images to visualize time-frequency components. In data preprocess, the lower frequency components, the more important features. Thus, the raw spectrogram images are filtered the components in the higher frequency. The dataset is split into 80% for training and validation and 20% for testing. The proposed lightweight CNN model, and pretrained VGG16 and Resnet50 models are trained using the training subset, with the validation subset preventing overfitting. The models’ performances are then evaluated using the testing subset, ensuring accuracy and reliability in the classification of blood-flow sounds.

  1. Line 87-98, the reviewed work of AI in edge computing can be removed regarding that the description (line 70-86) already focused on CNN researches in detecting auscultatory sounds for predicting CKD.

ANS: We remove these sentences.

Reviewer 2 Report

Comments and Suggestions for Authors

Please describe the number of training datasets required for CNN analysis. How will the subject number affect the training and validation accuracy?

Author Response

To Reviewer #2:

Thank the reviewer for his/her valuable comments that make better this manuscript. The texts in this revised manuscript have been corrected/ modified by red mark. It is our sincere hope that this revision will enhance readability and strengthen of the manuscript to satisfy the requirements of this prestigious journal.

Comments and Suggestions for Authors

Please describe the number of training datasets required for CNN analysis. How will the subject number affect the training and validation accuracy?

ANS: Many thanks for reviewer’s comment. We mention the number of dataset in Table 1. Than, we used five-fold cross-validation to assess the performance of models.

Line: 198-208

2.4. Dataset partitions

To train and test the model, the dataset must be divided. Table 1 shows the dataset partitions. The training dataset, making up 80% of the total, includes both the training and validation sets. The training set is used to fit the model, aiming for generalization, while the validation set, 20% of the training data, helps in model verification and parameter tuning. We ensure consistent data partitioning across experiments by fixing the random seed with the scikit-learn library. The test dataset, the remaining 20%, evaluates the model's accuracy on new data. Only the training and validation sets undergo data enhancement to improve model fit without affecting the authenticity of the test data, which includes 765 samples, with 385 for obstructed samples and 380 for unobstructed samples.

Table 1. Dataset partitions for obstructed and unobstructed classes.

Dataset

Label

Train

Test

Total

Raw signal

Obstructed class

1538

385

1923

Unobstructed class

1516

380

1896

Reviewer 3 Report

Comments and Suggestions for Authors

The paper presents a promising approach to detecting vascular access stenosis using a lightweight CNN model and offers valuable insights, particularly for edge computing applications. However, there are some issues raised, such as the need for additional explanations in certain sections and the inclusion of comparative data for a fair evaluation. Addressing these concerns will significantly strengthen the manuscript.

Major revision

1.    Could you elaborate on how the model performs with varied datasets or noise, particularly considering the variability in clinical data? In addition, The References are relatively insufficient compared to the Introduction; please add more.

2.    How do you assess the generalizability of the model with the current dataset? How might it perform across different demographics or clinical settings?

3.     For Figure 1, "Prevalence rates of treated end-stage renal disease (ESRD) in several countries," please revise the x-axis label and make overall modifications to the figure as needed. Additionally, if the data is sourced from other literature, could you please add the corresponding references?

4.    In section 2.2, Data Preprocessing, you mentioned that the STFT frequency range is set to 0-2000Hz because it includes the highest frequencies for detecting blood flow abnormalities. However, this seems like a rather broad range for detecting such abnormalities, so additional explanation is needed to support this statement.

5.    In section 2.2, Data Preprocessing, you mentioned that the STFT frequency range is set to 0-2000Hz because it includes the most relevant frequencies for detecting blood flow abnormalities. However, this range seems quite broad for detecting such abnormalities. Could you please provide additional explanation to support this choice?

6.    In Table 1, "Dataset partitions," it seems that the number of datasets is small, yet the results appear to be robust; please add an explanation for this observation.

7.    In section 3, Experimental Results, Table 2 only presents the evaluation metrics for the Lightweight-CNN model using the Full Frequency Spectrogram. To make a fair comparison with the results in Table 3, it seems necessary to include the metric results for the ResNet50 and VGG16 models using the Full Frequency Spectrogram as well. Could you please add this data?

8.    Would you consider including a comparison with other state-of-the-art methods in the literature? How does your model compare to traditional diagnostic methods or other machine learning models?

9.    How do you assess the generalizability of the model with the current dataset? How might it perform across different demographics or clinical settings?

Minor revision

-Please add References to the contents of sections 2.6, Loss Function, and 2.7, Evaluation Metrics.

-Most of the figures in this manuscript have text that is difficult to read, and the x-axis labels are tilted; please revise these issues.

-Please ensure that the formatting of the References is consistent.

- There are some inconsistencies in the use of terminology, such as "lightweight CNN," "2D CNN," and "Light-CNN." Would you ensure that these terms are used consistently throughout the manuscript to avoid any potential confusion?

- The figures and tables are informative, but could you enhance the captions with more detailed explanations, particularly for Figure 7? Additionally, have all figures and tables been clearly referenced and integrated into the text?

- Some references appear to be outdated or not directly relevant to the current study. Would you consider updating these with more current studies, or removing those that do not directly contribute to the focus of the manuscript?

- The manuscript mentions IRB approval, but could you provide more details on how patient confidentiality was maintained and how data was anonymized? This would help ensure compliance with ethical research standards.

Comments on the Quality of English Language

The paper presents a promising approach to detecting vascular access stenosis using a lightweight CNN model and offers valuable insights, particularly for edge computing applications. However, there are some issues raised, such as the need for additional explanations in certain sections and the inclusion of comparative data for a fair evaluation. Addressing these concerns will significantly strengthen the manuscript.

Major revision

1.    Could you elaborate on how the model performs with varied datasets or noise, particularly considering the variability in clinical data? In addition, The References are relatively insufficient compared to the Introduction; please add more.

2.    How do you assess the generalizability of the model with the current dataset? How might it perform across different demographics or clinical settings?

3.     For Figure 1, "Prevalence rates of treated end-stage renal disease (ESRD) in several countries," please revise the x-axis label and make overall modifications to the figure as needed. Additionally, if the data is sourced from other literature, could you please add the corresponding references?

4.    In section 2.2, Data Preprocessing, you mentioned that the STFT frequency range is set to 0-2000Hz because it includes the highest frequencies for detecting blood flow abnormalities. However, this seems like a rather broad range for detecting such abnormalities, so additional explanation is needed to support this statement.

5.    In section 2.2, Data Preprocessing, you mentioned that the STFT frequency range is set to 0-2000Hz because it includes the most relevant frequencies for detecting blood flow abnormalities. However, this range seems quite broad for detecting such abnormalities. Could you please provide additional explanation to support this choice?

6.    In Table 1, "Dataset partitions," it seems that the number of datasets is small, yet the results appear to be robust; please add an explanation for this observation.

7.    In section 3, Experimental Results, Table 2 only presents the evaluation metrics for the Lightweight-CNN model using the Full Frequency Spectrogram. To make a fair comparison with the results in Table 3, it seems necessary to include the metric results for the ResNet50 and VGG16 models using the Full Frequency Spectrogram as well. Could you please add this data?

8.    Would you consider including a comparison with other state-of-the-art methods in the literature? How does your model compare to traditional diagnostic methods or other machine learning models?

9.    How do you assess the generalizability of the model with the current dataset? How might it perform across different demographics or clinical settings?

Minor revision

-Please add References to the contents of sections 2.6, Loss Function, and 2.7, Evaluation Metrics.

-Most of the figures in this manuscript have text that is difficult to read, and the x-axis labels are tilted; please revise these issues.

-Please ensure that the formatting of the References is consistent.

- There are some inconsistencies in the use of terminology, such as "lightweight CNN," "2D CNN," and "Light-CNN." Would you ensure that these terms are used consistently throughout the manuscript to avoid any potential confusion?

- The figures and tables are informative, but could you enhance the captions with more detailed explanations, particularly for Figure 7? Additionally, have all figures and tables been clearly referenced and integrated into the text?

- Some references appear to be outdated or not directly relevant to the current study. Would you consider updating these with more current studies, or removing those that do not directly contribute to the focus of the manuscript?

- The manuscript mentions IRB approval, but could you provide more details on how patient confidentiality was maintained and how data was anonymized? This would help ensure compliance with ethical research standards.

Author Response

To Reviewer #3:

Thank the reviewer for his/her valuable comments that make better this manuscript. The texts in this revised manuscript have been corrected/ modified by red mark. It is our sincere hope that this revision will enhance readability and strengthen of the manuscript to satisfy the requirements of this prestigious journal.

Comments and Suggestions for Authors

The paper presents a promising approach to detecting vascular access stenosis using a lightweight CNN model and offers valuable insights, particularly for edge computing applications. However, there are some issues raised, such as the need for additional explanations in certain sections and the inclusion of comparative data for a fair evaluation. Addressing these concerns will significantly strengthen the manuscript.

Major revision

  1. Could you elaborate on how the model performs with varied datasets or noise, particularly considering the variability in clinical data? In addition, The References are relatively insufficient compared to the Introduction; please add more.

ANS: Many thanks for reviewer’s comments. In this study, we only used the self-collected data from 119 patients to classify the obstructed and unobstructed classes of vascular accesses. The VPG signals only were filtered by IIR filters to remove the noise. Thus, the limitations of this study included only using the single dataset. We mentioned these limitations in Discussion chapter. We modified the materials of Introduction, and deleted some sentences being relatively insufficient with the goal of this study. 

Line 435-441

This study has several limitations. The dataset, while robust, may not fully capture the diversity of patient conditions and blood-flow sound variations seen in one clinical practice. The transformation of audio signals into spectrogram images might introduce some information loss, affecting model performance. Additionally, the model's real-world applicability requires further validation in diverse clinical settings. Future research should address these issues by using more varied datasets, exploring alternative data representations, and conducting extensive clinical testing.

Line 68-86

Vascular phonoangiography (PAG) is a convenient technique to register the blood flow sounds through auscultation. The recorded sound signals have been analyzed with the time- and frequency-domain characteristics to evaluate the conditions of vascular access [11–15]. Moreover, various nonlinear analysis techniques have been applied to examine the irregularities in PAG signals of vascular access under both obstructed and unobstructed conditions [16]. Ota et al. have developed a method to detect and classify arteriovenous fistula stenosis severity using the fistula's sounds, converted into spectrograms for analysis with Convolutional Neural Network (CNN) and CNN plus Gate Recursive Unit (GRU) / Long Short-Term Memory (LSTM) models [17]. While the method shows promising results in a five-item classification test, its accuracy on new data is still low. Julkaew et al. have employed adaptive deep learning (DL) techniques to predict the quality of vascular access in patients undergoing hemodialysis [18]. Peralta et al. have applied the XGBoost algorithm to predict the failure of AVF within three months among patients receiving dialysis in-center [19]. Wang et al. have employed a DL model for diagnosing renal artery stenosis, using multimodal fusion of ultrasound images, spectral waveforms, and clinical data [20]. Li and He [21] have introduced a neural network algorithm designed for diagnosing arterial stenosis. Nguyen et al. [22] employed a 1D-CNN to assess signal quality for effectively predicting real-time blood flow volume. Furthermore, previous studies have used machine learning models to predict CKD [23–26].

  1. Li, Z.; He, W. Stenosis Diagnosis Based on Peripheral Arterial and Artificial Neural Network. Netw. Model. Anal. Heal. Informatics Bioinforma. 2021, 10, 13, doi:10.1007/s13721-021-00290-x.
  2. Nguyen, D.H.; Chao, P.C.-P.; Shuai, H.-H.; Fang, Y.-W.; Lin, B.S. Achieving High Accuracy in Predicting Blood Flow Volume at the Arteriovenous Fistulas of Hemodialysis Patients by Intelligent Quality Assessment on PPGs. IEEE Sens. J. 2022, 22, 5844–5856, doi:10.1109/JSEN.2022.3148415.

  1. How do you assess the generalizability of the model with the current dataset? How might it perform across different demographics or clinical settings?

ANS: Many thanks for reviewer’s comment. We only used the self-collected dataset to perform the five-fold cross-validation for assessing the generalizability of model. Thus, we mentioned this issue in the limitations of study.   

  1. For Figure 1, "Prevalence rates of treated end-stage renal disease (ESRD) in several countries," please revise the x-axis label and make overall modifications to the figure as needed. Additionally, if the data is sourced from other literature, could you please add the corresponding references?

ANS: We modify Figure. 1, and its mention is in Introduction chapter. The data is sourced from Refer [5].

Line 52-67

The number of patients with kidney diseases is rising in several countries. Especially, Taiwan has the highest prevalence of ESRD globally, surpassing both the general prevalence (the proportion of the population affected by the disease) and its incidence rate (the number of new patients relative to the total population) [5,6]. Furthermore, prior research indicates that the prevalence of chronic kidney disease (CKD) in Taiwan is alarmingly high at 11.3%, meaning nearly 1 in every 10 individuals suffers from CKD [7–9]. Statistics from the National Health Insurance Agency reveal that over 85,000 people were undergoing kidney dialysis in Taiwan in 2020, with numbers continuing to rise. Consequently, Taiwan is often referred to as a "kidney dialysis island," with a prevalence rate over 3,772 dialysis patients per million people, the highest ESRD prevalence rate treated globally, as shown in Figure 1 [5]. Additionally, Taiwan sees over 470 new diagnoses of ESRD per million people each year, the fastest rate worldwide [10].

Figure 1. Prevalence rates of treated end-stage renal disease in several countries.

[5] United States Renal Data System Available online: https://usrds-adr.niddk.nih.gov/2022/end-stage-renal-disease/11-international-comparisons.

  1. In section 2.2, Data Preprocessing, you mentioned that the STFT frequency range is set to 0-2000Hz because it includes the highest frequencies for detecting blood flow abnormalities. However, this seems like a rather broad range for detecting such abnormalities, so additional explanation is needed to support this statement.

ANS: Many thanks for reviewer’s comment. We correct the mentions in section 2.2.

Line 183-190

We can find their difference happening on lower frequencies. Thus, this study filtered the high frequency of spectrogram from 2,001 Hz to 100,000 Hz, as the reduced spectrogram images encompassed the most relevant information, as shown in Figure 4(c) and (d). The choice of the 0-2000 Hz frequency range in the STFT was to capture relevant physiological signals related to blood-flow sound. The higher band could have the information of non blood-flow sound or noise [31]. This narrow band allows the model to learn from a more comprehensive spectrum, potentially enhancing its ability to detect various patterns associated with different abnormalities.  

  1. In section 2.2, Data Preprocessing, you mentioned that the STFT frequency range is set to 0-2000Hz because it includes the most relevant frequencies for detecting blood flow abnormalities. However, this range seems quite broad for detecting such abnormalities. Could you please provide additional explanation to support this choice?

ANS: Many thanks for reviewer’s comment. In Table 2, the result metrics of VGG16, ResNet50, and Lightweight CNN model with the spectrogram images with the full band, which are lower than the result metrics with the filtered spectrogram images. Thus, we used spectrogram images with the narrow band to classify the stenosis of vascular access. This is also the contribution of this study.

Line 265-273

The performances of three different CNN models in classification using spectrogram images with the full band are provided in Table 2. In a two-class classification task, VGG16 model achieved an overall accuracy of 0.53, with strong recall (0.88) for obstructed vascular accesses but weaker performance for the unblocked class, reflected in lower precision (0.58) and recall (0.17). The ResNet50 model performed better overall, with a 0.59 accuracy, showing more balanced precision and recall for both classes, leading to a more consistent F1-score. The lightweight CNN model demonstrates a classification accuracy of 0.55 for the obstructed class, with precision and recall values of 0.58 and 0.38, respectively, resulting in an F1 score of 0.46.

  1. In Table 1, "Dataset partitions," it seems that the number of datasets is small, yet the results appear to be robust; please add an explanation for this observation.

ANS: Many thanks reviewer’s comment. We discuss this question in Discussion chapter.

Line 381-396

This study aims to develop a lightweight CNN model for the non-invasive detection of vascular access stenosis using blood-flow sounds, providing a novel approach to enhancing patient care for that undergoing hemodialysis. The proposed model exhibits rigorously compared with established CNN models, specifically VGG16 and ResNet50. To classify the obstructed and unobstructed fistula, signals of blood-flow sound are transformed into spectrogram images, a critical step that enables the extraction of relevant features from the time and frequency domains. We found that the low-frequency band has the more information for the blood-flow sound. In Table 2 and 3, the results metrics with the full band (0-10000 Hz) are lower than metrics with the narrow band (0-2000 Hz). Although, the number of dataset in Table 1 is small, the features of obstructed and unobstructed classes are very different. Thus, the models with the five-fold cross-validation have the well robust and performance in Table4, 5, and 6. The results, as presented in Table 3, demonstrate that the lightweight CNN model achieves an unparalleled prediction accuracy of 100%, significantly outperforming ResNet50 and VGG16, which record accuracies of 99% and 95%, respectively. This superior performance underscores the efficacy of the lightweight CNN model in accurately identifying vascular conditions.

  1. In section 3, Experimental Results, Table 2 only presents the evaluation metrics for the Lightweight-CNN model using the Full Frequency Spectrogram. To make a fair comparison with the results in Table 3, it seems necessary to include the metric results for the ResNet50 and VGG16 models using the Full Frequency Spectrogram as well. Could you please add this data?

ANS: Many thanks reviewer’s comment. We modify the result metrics in Table 2, and the fusion matrixes in Figure 6.

Table 2. Evaluation results matrix for the VGG16, ResNet50 and lightweight CNN models using spectrogram images with the full band.

Models

Accuracy

Obstructed Class

Unobstructed Class

Precision

Recall

f1-score

Precision

Recall

f1-score

VGG16

0.53

0.52

0.88

0.65

0.58

0.17

0.26

ResNet50

0.59

0.59

0.62

0.60

0.59

0.56

0.57

Light-CNN

0.55

0.58

0.38

0.46

0.53

0.72

0.61

(a)

(b)

(c)

Figure 6. Confusion matrixes for the different CNN models using spectrogram images with the full band, (a) VGG16; (b) ResNet50; (c) Lightweight CNN.

  1. Would you consider including a comparison with other state-of-the-art methods in the literature? How does your model compare to traditional diagnostic methods or other machine learning models?

ANS: Many thanks for reviewer’s comments. We added some sentences to compare the previous studies in Discussions chapter.

Line 397-411

Mansy et al. studied 11 patients whose vascular sounds were recorded before and after angiography. They found that changes in acoustic amplitude and/or spectral energy distribution. Certain acoustic parameters with the change in the degree of stenosis had the high correlation coefficient, approaching to 0.98 [35]. Sung et al, proposed a multivariate Gaussian distribution (MGD) model to detect significant vascular access stenosis with the phonoangiographic method. The auditory spectrum flux and auditory spectral centroid were extracted from the blood-flow sound of 16 hemodialysis patients with AVG. its accuracy only approached to 83.9% [12]. In this study, we complied 119 patient records. There were 106 datasets available belonging to the stenosis or obstructed case, and 105 datasets belonging to the unobstructed case. The signal was transferred to a spectrogram image by STFT. In order to enhance the feature of spectrogram image, the filtered spectrogram image was used to classify the obstructed and unobstructed classes with the proposed lightweight CNN model. There is a 100% of accuracy. This result shows that the two-dimension CNN model could have the potential benefit to detect the vascular access stenosis. 

  1. How do you assess the generalizability of the model with the current dataset? How might it perform across different demographics or clinical settings?

ANS: Many thanks for reviewer’s comment. We only used the self-collected dataset to perform the five-fold cross-validation for assessing the generalizability of model. Thus, we mentioned this issue in the limitations of study.

Minor revision

  1. Please add References to the contents of sections 2.6, Loss Function, and 2.7, Evaluation Metrics.

ANS: Many thanks for reviewer’s comment. We added some references in section 2,6 and 2.7.

Line 232-239

2.6. Loss Function

In each instance of training, we quantify the discrepancy between the predicted and actual data. This quantification serves as a crucial indicator of the model's efficacy, encapsulated by the loss function. The present investigation adopts a binary classification framework, wherein the output layer's activation function is delineated by a Sigmoid curve. Consequently, the output values are confined within the interval [0, 1]. Given this constraint, the binary cross-entropy function emerges as the appropriate choice for evaluating model performance [33]. The following is the formula for binary cross entropy:

Line: 247-252

2.7. Evaluation Metrics

This study analyzed the model's performance using precision, recall, specificity, negative predictive value, and accuracy [34]. In the equations below, 'TP', 'TN', 'FP', and 'FN' are used to represent 'True Positive' (accurate positive detections), 'True Negative' (correct negative classifications), 'False Positive' (incorrect positive classifications), and 'False Negative' (incorrect negative classifications), respectively.

  1. Most of the figures in this manuscript have text that is difficult to read, and the x-axis labels are tilted; please revise these issues.

ANS: Many thanks for reviewer’s comment. We modify these sentences and labels of figures.

  1. Please ensure that the formatting of the References is consistent.

ANS: Many thanks for reviewer’s comment. We modify the format of Ref. [7].

  1. There are some inconsistencies in the use of terminology, such as "lightweight CNN," "2D CNN," and "Light-CNN." Would you ensure that these terms are used consistently throughout the manuscript to avoid any potential confusion?

ANS: Many thanks for reviewer’s comment. We correct as “lightweight CNN model”.

  1. The figures and tables are informative, but could you enhance the captions with more detailed explanations, particularly for Figure 7? Additionally, have all figures and tables been clearly referenced and integrated into the text?

ANS: Many thanks for reviewer’s comment. We modify the captions of all tables and figures, and mention them in text. Figure 7 also is modified.

(a)

(b)

(c)

Figure 7. Confusion matrixes for the different CNN models using filtered spectrogram images, (a) VGG16; (b) ResNet50; (c) lightweight CNN

  1. Some references appear to be outdated or not directly relevant to the current study. Would you consider updating these with more current studies, or removing those that do not directly contribute to the focus of the manuscript?

ANS: Many thanks for reviewer’s comment. We delete nonrelative references and add some references as Ref. [21,22,27,28,29,30,31,33,34].

  1. Li, Z.; He, W. Stenosis Diagnosis Based on Peripheral Arterial and Artificial Neural Network. Netw. Model. Anal. Heal. Informatics Bioinforma. 2021, 10, 13, doi:10.1007/s13721-021-00290-x.
  2. Nguyen, D.H.; Chao, P.C.-P.; Shuai, H.-H.; Fang, Y.-W.; Lin, B.S. Achieving High Accuracy in Predicting Blood Flow Volume at the Arteriovenous Fistulas of Hemodialysis Patients by Intelligent Quality Assessment on PPGs. IEEE Sens. J. 2022, 22, 5844–5856, doi:10.1109/JSEN.2022.3148415.
  3. Hakim, R.; Himmelfarb, J. Hemodialysis Access Failure: A Call to Action. Kidney Int. 1998, 54, 1029–1040, doi:10.1046/j.1523-1755.1998.00122.x.
  4. Hoeben, H.; Abu-Alfa, A.K.; Reilly, R.F.; Aruny, J.E.; Bouman, K.; Perazella, M.A. Vascular Access Surveillance: Evaluation of Combining Dynamic Venous Pressure and Vascular Access Blood Flow Measurements. Am. J. Nephrol. 2003, 23, 403–408, doi:10.1159/000074297.
  5. Jeon, H.; Jung, Y.; Lee, S.; Jung, Y. Area-Efficient Short-Time Fourier Transform Processor for Time–Frequency Analysis of Non-Stationary Signals. Appl. Sci. 2020, 10, 7208, doi:10.3390/app10207208.
  6. Mateo, C.; Talavera, J.A. Short-Time Fourier Transform with the Window Size Fixed in the Frequency Domain. Digit. Signal Process. 2018, 77, 13–21, doi:10.1016/j.dsp.2017.11.003.
  7. Rao, A.; Huynh, E.; Royston, T.J.; Kornblith, A.; Roy, S. Acoustic Methods for Pulmonary Diagnosis. IEEE Rev. Biomed. Eng. 2019, 12, 221–239, doi:10.1109/RBME.2018.2874353.
  8. Yu, R.; Wang, Y.; Zou, Z.; Wang, L. Convolutional Neural Networks with Refined Loss Functions for the Real-Time Crash Risk Analysis. Transp. Res. Part C Emerg. Technol. 2020, 119, 102740, doi:10.1016/j.trc.2020.102740.
  9. Dj Novakovi, J.; Veljovi, A.; Ili, S.S.; Zeljko Papi, ˇ; Tomovi, M. Evaluation of Classification Models in Machine Learning. Theory Appl. Math. Comput. Sci. 2017, 7, 39–46.

  1. The manuscript mentions IRB approval, but could you provide more details on how patient confidentiality was maintained and how data was anonymized? This would help ensure compliance with ethical research standards.

ANS: Many thanks for reviewer’s comment. We add some sentences to mention the procedure of experiment.

Line: 109-122

  1. Materials and Methods

The flowchart of this study for identifying whether the stenosis or obstruction of vascular access or not with the blood-flow sound is illustrated in Figure 2. The blood-flow sound signals were collected from patients at the Cardiovascular Center of the E-Da Hospital. The physician identified the vascular access of patients with the stenosis condition, and performed the angioplasty surgery to clear the obstructed blood-flow pathway. Then, the blood-flow sounds are measured before and after the surgery, which are converted into spectrogram images to visualize time-frequency components. In data preprocess, the lower frequency components, the more important features. Thus, the raw spectrogram images are filtered the components in the higher frequency. The dataset is split into 80% for training and validation and 20% for testing. The proposed lightweight CNN model, and pretrained VGG16 and Resnet50 models are trained using the training subset, with the validation subset preventing overfitting. The models’ performances are then evaluated using the testing subset, ensuring accuracy and reliability in the classification of blood-flow sounds.

Line: 126-131

2.1. Data Source

This clinical trial was approved by the Institutional Review Board of the E-Da Hospital, Kaohsiung, Taiwan (No. EMRP-107-142). and informed consent was acquired from each participant prior to initiation of the study. In the informed consent, the physician explained the materials and measurement procedure of this experiment. Moreover, the blood-flow sound measurement was the noninvasive measurement.

Reviewer 4 Report

Comments and Suggestions for Authors

Well written and presented. Looking forward to application of this technology to other fields of medicine.

However you need to remove the placeholder text "This section may be divided by subheadings. It should provide a concise and precise description of the experimental results, their interpretation, as well as the experimental conclusions that can be drawn." from section 3.

Also... I suggest explaining what Fourier transformation is for medical professionals that might be interested in this research but are not technically / mathematically inclined.

Author Response

To Reviewer #4:

Thank the reviewer for his/her valuable comments that make better this manuscript. The texts in this revised manuscript have been corrected/ modified by red mark. It is our sincere hope that this revision will enhance readability and strengthen of the manuscript to satisfy the requirements of this prestigious journal.

Comments and Suggestions for Authors

Well written and presented. Looking forward to application of this technology to other fields of medicine.

  1. However you need to remove the placeholder text "This section may be divided by subheadings. It should provide a concise and precise description of the experimental results, their interpretation, as well as the experimental conclusions that can be drawn." from section 3.

ANS: We removed this sentence.

Line 264-279

  1. Experimental Results

3.1     Classification Result of CNN Models

The performances of three different CNN models in classification using spectrogram images with the full band are provided in Table 2. In a two-class classification task, VGG16 model achieved an overall accuracy of 0.53, with strong recall (0.88) for obstructed vascular accesses but weaker performance for the unblocked class, reflected in lower precision (0.58) and recall (0.17). The ResNet50 model performed better overall, with a 0.59 accuracy, showing more balanced precision and recall for both classes, leading to a more consistent F1-score. The lightweight CNN model demonstrates a classification accuracy of 0.55 for the obstructed class, with precision and recall values of 0.58 and 0.38, respectively, resulting in an F1 score of 0.46. These metrics indicate moderate precision but low recall, meaning the model is accurate in predicting obstructed class but misses many actual obstructed cases. The unobstructed class achieved a recall of 0.72, higher than its precision measure of 0.53 and f1-score of 0.61. The higher precision suggests that when the model predicts the unobstructed class, it is more often correct; better recall means it is good at identifying actual cases of unobstructed class.

  1. Also... I suggest explaining what Fourier transformation is for medical professionals that might be interested in this research but are not technically / mathematically inclined.

ANS: We added some sentences to mention STFT in sector 2.2.

Line 167-174

2.2. Data Preprocessing

The STFT is introduced as a method to address the inherent limitations of the fast Fourier transform (FFT), particularly in the analysis of non-stationary or noisy signals. STFT is predominantly utilized for the extraction of narrow-band frequency components within such signals [29]. The underlying concept involves segmenting the original signal into small, localized time windows, followed by the application of FFT to each segment. This approach enables the representation of temporal variations in the signal's frequency content, effectively capturing the dynamic behavior within each time window [30].

Round 2

Reviewer 1 Report

Comments and Suggestions for Authors

Thank you to all the authors for their contributions! Could you please provide a table listing the state-of-the-art methods in this field? Based on this, it would be helpful to emphasize the novelty of the proposed approach.

Author Response

To Reviewer #1:

Thank the reviewer for his/her valuable comments that make better this manuscript. The texts in this revised manuscript have been corrected/ modified by red mark. It is our sincere hope that this revision will enhance readability and strengthen of the manuscript to satisfy the requirements of this prestigious journal.

Thank you to all the authors for their contributions! Could you please provide a table listing the state-of-the-art methods in this field? Based on this, it would be helpful to emphasize the novelty of the proposed approach.

ANS: Many thanks for reviewer’s comment. We modified some sentences to mention the novelty of proposed method in Discussion chapter, and added Table 8 to mention the state-of -art methods in this field.

Line 382-443

  1. Discussion

This study aims to develop a lightweight CNN model for the non-invasive detection of vascular access stenosis using blood-flow sounds, providing a novel approach to enhancing patient care for that undergoing hemodialysis. The proposed model exhibits rigorously compared with established CNN models, specifically VGG16 and ResNet50. To classify the obstructed and unobstructed fistula, signals of blood-flow sound are transformed into spectrogram images, a critical step that enables the extraction of relevant features from the time and frequency domains. In this study, we used spectrogram images with the narrow bandwidth, below 2000 Hz because we observed that the low-frequency band carries more significant information related to blood-flow sounds. This finding aligns with prior research, as the selected frequency range was informed by earlier studies [35]. In Table 2 and 3, the results metrics with the full band (0-10000 Hz) are lower than metrics with the narrow band (0-2000 Hz). Notably, the performances of all models were consistently higher when using the spectrogram images with the narrow band. This finding also showed that the bandwidth of blood flow sound would be at the low frequency.  Filtering out higher frequencies not only reduced noises but also emphasized relevant features, and enhancing the model’s performance. Although, the number of samples in Table 2 is small, the features of obstructed and unobstructed classes are very different. Thus, the models with the five-fold cross-validation have the well robust and performance in Table 4, 5, and 6. The results, as presented in Table 3, demonstrate that the lightweight CNN model achieves an unparalleled prediction accuracy of 100%, significantly outperforming ResNet50 and VGG16, which record accuracies of 99% and 95%, respectively. This superior performance underscores the efficacy of the lightweight CNN model in accurately identifying vascular conditions.

Mansy et al. studied 11 patients whose vascular sounds were recorded before and after angiography. They found that changes in acoustic amplitude and/or spectral energy distribution. Certain acoustic parameters with the change in the degree of stenosis had the high correlation coefficient, approaching to 0.98 [36]. Sung et al, proposed a multivariate Gaussian distribution (MGD) model to detect significant vascular access stenosis with the phonoangiographic method. The auditory spectrum flux and auditory spectral centroid were extracted from the blood-flow sound of 16 HD patients with AVG. its accuracy only approached to 83.9% [12]. In this study, we complied 119 patient records. There were 106 datasets available belonging to the stenosis or obstructed case, and 105 datasets belonging to the unobstructed case. The signal was transferred to a spectrogram image by STFT. In order to enhance the feature of spectrogram image, the filtered spectrogram image was used to classify the obstructed and unobstructed classes with the proposed lightweight CNN model. There is a 100% of accuracy. This result shows that the two-dimension CNN model could have the potential benefit to detect the vascular access stenosis. 

To assess the suitability of the model for edge computing applications, the study also analyzes the memory size. The lightweight CNN model is identified as having the smallest memory size compared to VGG16 and ResNet50, making it especially suitable for use in environments with limited resources. Its compact memory size allows it to function efficiently on devices with restricted computational power and storage capacity, such as mobile devices or portable medical equipment. The memory size of model is a crucial aspect of edge computing in healthcare, as it determines the feasibility of deploying advanced algorithms on resource-constrained devices [37]. This efficiency is crucial in real-time medical applications where rapid and accurate predictions are necessary for timely interventions.

This study also compares our results with previous edge computing study, by Zhou et al. [38], where they have achieved a specificity value of 0.791. In contrast, our study demonstrates a specificity value exceeding 95% across all employed models. Moreover, another study Ota et al. [17] have achieved an accuracy of 0.821, precision of 0.414, and recall of 0.578, whereas our study outperforms these results. This indicates a significant improvement in the ability to detect the vascular access stenosis. Table 8 summarizes various studies that employed deep learning models to analyze AVF sounds and clinical data for predicting AVF functionality and vascular access quality in HD patients. The dataset, model type, methods used, and key outcomes are outlined for each study, highlighting the performance and predictive accuracy of the models.

Table 8. Summary of deep learning models for predicting AVF functionality and VA quality in HD patients.

Author

Dataset

Model

Method

Outcome

Ota et al. [17]

AVF sounds recording 1 minute from 20 patients.

CNN + BiLSTM (Bidirectional Long Short-Term Memory)

Extracted heartbeat-specific arteriovenous fistula sounds.

0.70% to 0.93 of Accuracy, 0.75 to 0.92 of AUC.

Julkaew et al. [18]

Clinical data from 398 HD patients

DeepVAQ -CNN

Used Photoplethysmogram (PPG) to predict VA quality, trained and fine-tuned CNN.

0.92 of Accuracy, 0.96 of Specificity, 0.88 of Precision, 0.84 of F1 Score, 0.86 of AUC.

Peralta et al. [19]

13,369 dialysis patients (EuCliD® Database)

AVF-FM (XGBoost)

Predict AVF failure in 3 months.

0.80 of AUC (95% CI 0.79–0.81)

Nguyen et al. [22]

300 qualified and 202 unqualified PPG waveforms

1D-CNN +

FCNN

Developed an ML algorithm to assess PPG signal quality for prediction of blood flow volume, using waveform quality criteria.

Transformed NN: 0.94 of Accuracy.

1D-CNN: 0.95 of Accuracy.

Zhou et al. [38]

2,565 AVF blood flow sounds from 433 patients

Vision Transformer (ViT)

AVF sounds from 6 locations, pre-processed into Mel-spectrograms and recurrence plots.

ViT: 0.92 of Sensitivity, 0.79 of  Specificity, 0.91 of F1 Score

Chung et al. [23]

437 audio recordings from 84 HD patients

CNN, ViT-GRU

AV access bruit recordings converted to Mel-spectrograms. Models trained to predict dysfunction.

CNN: 0.70 of F1 Score, 0.71 of AUC.

ViT-GRU: 0.52 of F1 Score, 0.60 of AUC.

Park et al. [39]

80 audio files  from 40 HD patients.

ResNet50, EfficientNetB5, DenseNet201

Digital AVF sounds recorded, converted to mel spectrograms, and used DCNN models.

ResNet50: 0.99 of AUC.

EfficientNetB5: 0.98 of AUC.

Reviewer 3 Report

Comments and Suggestions for Authors

Thank you for addressing my previous inquiries. However, there are still several aspects of the document that require further refinement and elaboration.

Major Revision1. In Section 2. Materials and Methods, please provide a detailed explanation of why lower frequencies are considered more important for this study.2. In Section 2.2. Data Preprocessing, the choice of the frequency range 0-2000Hz seems to be based on the statement from reference [31]: "The frequency range considered physiologically important for nearly all heart and lung sounds is up to 2000 Hz [6; 78]." It seems that specifying the range as 0-2000Hz might be too broad. Moreover, since this paper is focused on acoustic methods for lung diagnosis rather than blood flow sounds, it would be advisable to consult additional references specific to blood flow sounds.Minor Revision1. Figure 2: Please enhance the quality of the image. The current figure appears to lack detail and clarity.2. Figure 3: The text colors for Artery and Vein seem to clash with the background image. 3. Figures 6 and 7: Modify the y-axis labels in these figures to match the format of the y-axis label in Figure 4.

Comments on the Quality of English Language

None.

Author Response

To Reviewer #3:

Thank the reviewer for his/her valuable comments that make better this manuscript. The texts in this revised manuscript have been corrected/ modified by red mark. It is our sincere hope that this revision will enhance readability and strengthen of the manuscript to satisfy the requirements of this prestigious journal.

Major Revision

  1. In Section 2. Materials and Methods, please provide a detailed explanation of why lower frequencies are considered more important for this study.2. In Section 2.2. Data Preprocessing, the choice of the frequency range 0-2000Hz seems to be based on the statement from reference [31]: "The frequency range considered physiologically important for nearly all heart and lung sounds is up to 2000 Hz [6; 78]." It seems that specifying the range as 0-2000Hz might be too broad. Moreover, since this paper is focused on acoustic methods for lung diagnosis rather than blood flow sounds, it would be advisable to consult additional references specific to blood flow sounds.

ANS: Many thanks for reviewer’s comment. We removed the sentences about Reference [31], and added some sentences in Discussion chapter to explain why we have this inspiration to choose the bandwidth of 2000 Hz from Ref. [35].   

Line 382-405

  1. Discussion

This study aims to develop a lightweight CNN model for the non-invasive detection of vascular access stenosis using blood-flow sounds, providing a novel approach to enhancing patient care for that undergoing hemodialysis. The proposed model exhibits rigorously compared with established CNN models, specifically VGG16 and ResNet50. To classify the obstructed and unobstructed fistula, signals of blood-flow sound are transformed into spectrogram images, a critical step that enables the extraction of relevant features from the time and frequency domains. In this study, we applied a frequency filter to isolate the range between 0 and 2000 Hz and observed that the low-frequency band carries more significant information related to blood-flow sounds. This finding aligns with prior research, as the selected frequency range was informed by earlier studies [35]. The decision to emphasize lower frequencies is based on the hypothesis that this range may more effectively capture the hemodynamic characteristics of blood flow. In Table 2 and 3, the results metrics with the full band (0-10000 Hz) are lower than metrics with the narrow band (0-2000 Hz). Notably, the model's performance metrics were consistently higher when using the narrow frequency band. This finding suggests that filtering out higher frequencies not only reduces noise but also emphasizes relevant features, and enhancing the model performance. Although, the number of samples in Table 2 is small, the features of obstructed and unobstructed classes are very different. Thus, the models with the five-fold cross-validation have the well robust and performance in Table 4, 5, and 6. The results, as presented in Table 3, demonstrate that the lightweight CNN model achieves an unparalleled prediction accuracy of 100%, significantly outperforming ResNet50 and VGG16, which record accuracies of 99% and 95%, respectively. This superior performance underscores the efficacy of the lightweight CNN model in accurately identifying vascular conditions.

  1. Kurokawa, Y.; Abiko, S.; Watanabe, K. Noninvasive Detection of Intracranial Vascular Lesions by Recording Blood Flow Sounds. Stroke 1994, 25, 397–402, doi:10.1161/01.STR.25.2.397.

Minor Revision

  1. Figure 2: Please enhance the quality of the image. The current figure appears to lack detail and clarity.

ANS: Many thanks for reviewer’s comment. We improved the image quality of Fig.2.

  1. Figure 3: The text colors for Artery and Vein seem to clash with the background image.

ANS: Many thanks for reviewer’s comment. We changed the color of the words of “artery” and “vein” to black in Fig.3.

  1. Figures 6 and 7: Modify the y-axis labels in these figures to match the format of the y-axis label in Figure 4.

ANS: Figure 4 represents the spectrogram image, and Figures 6 and 7 represent the confusion matrix. Since they are different images, we cannot use the same y-axis format for both.

Round 3

Reviewer 3 Report

Comments and Suggestions for Authors

He was well-organized and responded diligently to every question.

Comments on the Quality of English Language

He was well-organized and responded diligently to every question.